# Identifying clinico-radiological determinants of post-stroke fatigue 3 months post-stroke in a French hospital-based cohort of non-severe stroke patients without psychiatric comorbidities

**Suhrit Duttagupta**[1,*☉], **Lutzi Castano**[1,2☉], **Sandra Chanraud**[1,2], **Igor Sibon**[1,3], **Sylvie Berthoz**[1,4]

**1** University of Bordeaux, INCIA CNRS UMR 5287, Bordeaux, France, **2** EPHE PSL Research University, Life and Earth Sciences, Paris, France, **3** CHU Bordeaux, Unité Neurovasculaire, Bordeaux, France, **4** Department of Psychiatry for Adolescents and Young Adults, Institut Médico-chirurgical Montsouris, Paris, France

☉ These authors contributed equally to this work.
\* suhrit.duttagupta@u-bordeaux.fr

## Abstract

Post-stroke fatigue (PSF) is an overlooked and debilitating condition. As a multidimensional construct, fatigue encompasses physical, cognitive, and emotional components, complicating efforts to understand PSF pathophysiological mechanisms and identify key predictors. We aimed to investigate the impact of lesion characteristics on different facets of subacute PSF while accounting for socio-demographic, psychological, and neurological factors. We assessed 231 patients with first-ever mild ischemic stroke without recent anxiety or depressive disorders using the Multidimensional Fatigue Inventory (MFI) at 3 months and the Hospital Anxiety and Depression Scale (HAD), alongside routine clinical evaluations. Lesion analysis was performed using two approaches: a voxel-based method using support vector regression-based multivariate lesion-symptom mapping (SVR-LSM), and a network-based method using principal component analysis (PCA) of lesioned gray and white matter regions. PSF had an overall prevalence of 20.8%, was more frequent in women and younger patients, and was associated with HAD scores. SVR-LSM identified associations between lesions in the right corona radiata and external capsule with total MFI scores, but not with HAD scores. After adjusting for relevant confounders, the network-based approach revealed associations between mental fatigue and reduced activity subdimensions and brain components involving cerebro-cerebellar tracts. Our findings indicate that, in a relatively homogeneous population, PSF arises from an interplay of socio-demographic, emotional, and cerebral risk factors. The involvement of motor pathways raises the possibility that neuronal overactivity, compensating for disrupted networks, may contribute to long-term fatigue. Further studies in more

**Data availability statement:** Data cannot be shared publicly because of sensitive patient information, adhering to the French law no. 2012-300 of 5 March 2012 on research involving the human person and as authorized by the ethics committee, Comité de protection de personnes (CPP) Sud-Est III. Upon reasonable request, anonymized numerical data may be provided to researchers meeting the criteria for access to confidential data from the Methodological Support for Clinical and Epidemiological Research (USMR) of Bordeaux University Hospital. Requests must be sent to the USMR chief Dr Antoine BENARD - antoine.benard@u-bordeaux.fr.

**Funding:** Funders of this work were the French ministry of Health (reference number: PHRC-17-0377) and the French National Research Agency (ANR), under the France 2030 program (reference number: ANR-23-IAHU-0001). Funders of the PhD students involved in this work are the Doctoral School of Bordeaux Neurocampus Graduate Program (SD) and the EPHE-PSL Research University (LC).

**Competing interests:** The authors have declared that no competing interests exist.

diverse populations along with whole-brain analyses would validate the generalizability of our results.

## Introduction

Stroke is one of the main causes of reduced disability-adjusted life years (DALYs) worldwide [1]. In addition to physical disability, frequent but less apparent complications such as fatigue, emotional problems, and cognitive impairments significantly influence post-stroke outcome. Post-Stroke Fatigue (PSF) is commonly defined as "a feeling of exhaustion, weariness or lack of energy that can be overwhelming, and which can involve physical, emotional, cognitive and perceptual contributors, which is not relieved by rest and affects a person's daily life" [2]. PSF is a frequent, long-lasting invisible condition with an estimated overall prevalence of 45.2% [3].

Although pre-stroke fatigue, sleep disorders, and stroke severity have been established as prominent PSF risk factors, several others remain to be considered: the most recent meta-analysis [3] found a robust association between PSF and age, but not female sex, contradicting previous work [4]. In the literature on PSF, variability in the sample characteristics, such as sex ratio and stroke severity, may cause discrepancy in findings across studies [4].

Furthermore, the potential impact of comorbid physical and mental health conditions on PSF is often insufficiently addressed. In particular, fatigue and depressive mood frequently co-occur and share partially overlapping experiential dimensions such as reduced energy and psychomotor slowing, making it challenging to disentangle their unique contributions [5]. Yet, too few studies explicitly model depressive symptoms – or broader negative mood states – as confounders when examining PSF. This omission limits the interpretability and generalizability of reported associations.

It has also been posited that evaluations for measuring fatigue may be too simplistic to detect associations with neuroimaging factors. The most commonly used instrument for evaluating PSF is the unidimensional Fatigue Severity Scale (FSS). While a recent international roundtable ranked the FSS highest for desirable psychometric properties such as construct validity and reliability [2], it is now widely recognized that fatigue is a multidimensional construct that includes physical, cognitive, motivational, and affective components [6]. Reliance on a predominantly unidimensional measure may therefore obscure clinically relevant subdomains of PSF and lead to specific fatigue profiles being underrecognized. In contrast, the Multidimensional Fatigue Inventory (MFI) provides structured subscales that capture the different aforementioned forms of fatigue. Importantly, the MFI was judged in the aforementioned roundtable to be comparable to the FSS in terms of psychometric adequacy for inferential statistical analysis, while offering the additional advantage of domain-level resolution. Given the need to identify distinct PSF subcomponents and their respective correlates, the MFI represents a more suitable tool for multidimensional investigation.

In terms of brain lesion markers, recent meta-analyses found no robust association with markers of cerebral small vessel disease such as the extent of white matter

hyperintensities or number of microbleeds [7]. Considering stroke lesion characteristics, a higher prevalence of PSF following intracerebral hemorrhage was reported, but there was no robust evidence for an association between PSF and lesion location, lateralization, or volume [4,8]. Although lesions in the brainstem and thalamus have been highlighted, it is difficult to draw firm conclusions due to substantial methodological heterogeneity [4]. Specifically, studies vary on the level of analysis: from voxel-based, regional, or network-based. Considering the inconsistent findings across these analyses, applying them based on the same cohort would provide evidence for their complementarity.

With the current status of the literature, the present study used a multidimensional fatigue screening tool to investigate risk factors of PSF. To specifically examine stroke-related mechanisms, we recruited participants with minor ischemic strokes and no recent thymic disorders while taking potential confounding factors across physical, psychological, and clinical domains. To determine neuroanatomical risk factors, we performed lesion-symptom mapping through a voxel-based and a network-based approach.

## Materials and methods

### Participants

The study participants were part of a larger prospective cohort recruited in Bordeaux, France (Protocol: MOTIVPOSDEP, Clinical Trials: NCT04043052) from September 9, 2020 until September 26, 2024. The primary inclusion criteria were chosen to minimize confounding influences on post-stroke fatigue and mood, ensuring that any observed fatigue was attributable to the stroke itself rather than severe functional or psychiatric effects: recent ischemic stroke (≤15 days ago), no severe handicap (modified Rankin scale, mRS ≤ 3), no post-stroke severe cognitive impairment (Montreal Cognitive Assessment, MoCA ≥ 16) or severe aphasia (National Institute of Health Stroke Scale, NIHSS item 9 ≥ 2), discharged to home, no psychoactive drug use during the preceding month, and no current or recent (over the past 6 months) anxiety or depression disorders or severe substance use disorders (except tobacco) according to the Mini-International Neuropsychiatric Interview. The study was conducted in accordance with the Declaration of Helsinki and the Institutional Review Boards and local ethics committee (CPP Sud-Est III) approved the study protocol. Written informed consent to participate and for publication was obtained from all participants prior to inclusion. Capacity to provide consent was determined by the participant's ability to read and write, and the absence of legal protection, which was approved by the local ethics committee.

### Evaluations

Clinical assessments collected before discharge (T1; within 15 days of stroke) and at 3 months post-stroke (T2) included: the mRS and NIHSS (upon admission for T1) for neurological status, the Hospital Anxiety and Depression scale (HAD-A and HAD-D) for mood status, and the MoCA. Stroke risk factors examined were obesity, diabetes, hypertension, dyslipidemia, alcohol consumption and smoking dependency (Heavy Smoking Index, HSI). For both HAD subscales, a score > 7 was used to define groups with anxiety or depression. Missing data (seven HAD-A values at T1 and one at T2) were not imputed.

PSF was assessed only at T2 using the Multidimensional Fatigue Inventory (MFI) [6], a 20-item self-report questionnaire including five subscales: General Fatigue, Physical Fatigue, Reduced Motivation, Reduced Activity, and Mental Fatigue, where higher scores reflect greater fatigue. For the MFI total score, a cut-off of 60 was used and the following cut-offs were used for the subscores: 11 for General fatigue, 10 for Reduced activity, and 9 for Physical fatigue, Reduced motivation and Mental fatigue [9]. Based on these cut-offs, patients were classified into fatigue-present and fatigue-absent groups, hereafter referred to as fatigue status groups for categorical analysis, in addition to the scores being used for dimensional analysis.

### Image acquisition

Clinical MRI scans were acquired upon hospital admission with a SIEMENS Magnetom Aera 1.5 Tesla scanner. Acquisition parameters for the diffusion-weighted imaging (DWI) sequence were: number of directions 3, b-value 1000, TR

5000 ms, TE 78 ms, flip angle 90°, slice thickness 2 mm, slice gap 2.4 mm, voxel size 0.78 × 0.78 × 2.4 mm³, field of view 270 × 245 mm², pixel bandwidth 1455, and number of averages 8.

### Lesion preprocessing

Ischemic lesions were delineated from DWI images with the Acute-Stroke Detection Segmentation (ADS) toolbox [10]. Lesions were inspected individually and manually corrected if required using ITK-SNAP 3.8 (www.itksnap.org). Spatial normalization was performed with the BCB-toolkit (https://storage.googleapis.com/bcblabweb/index.html). Normalized lesions were resampled to a voxel size of 2 x 2 x 2 mm³ using trilinear interpolation. Lesion volumes were derived from normalized images to account for variability in brain volume.

### Statistical analyses

Descriptive statistics for all evaluations and population characteristics were reported, in the overall sample and then split by fatigue status groups. Categorical and dimensional analyses were reported separately, with univariate and multivariate results: for categorical analyses based on fatigue status groups, cross-sectional analyses were performed using the χ² test of independence, group comparisons were done using Mann-Whitney U tests – reporting effect sizes as rank biserial correlation ($r_{rb}$) – and multivariate comparisons were performed using ranked ANCOVA tests after ensuring there was no collinearity of the variables based on a VIF score < 5. For dimensional analyses between clinical factors and fatigue scores, Spearman's correlations and partial correlations were conducted. Additionally, stratified group comparisons were done by age – grouped as under 55, between 55–70, and over 70 years – and by sex. To identify confounding variables for multivariate analyses, associations between clinical factors were examined (S1 Table) and those that were significantly correlated with an MFI score as well as another clinical variable were selected. To verify the absence of confounding effects between fatigue and mood evaluations, the 14 items of the HAD and 20 items of the MFI scales were entered into an exploratory factor analysis with promax rotation and minimal residual extraction method. Statistical significance was set at $p < 0.05$ and a Bonferroni correction was applied to control for false discovery rate (FDR). All analyses were performed using SPSS version 21.0 (IBM Corp.).

For neuroimaging analyses, we initially implemented multivariate support vector regression lesion-symptom mapping (SVR-LSM). Segmented lesions of the patients were inputted in the DeMarco and Turkeltaub's toolbox [11] (on MATLAB_R2018a, relying on the machine learning library LibSVM and on SPM12 for image manipulation), along with the patients' MFI scores. The lesion overlap threshold was set as 5, with 1000 permutations and a clusterwise threshold of $p < 0.05$. With corrections based on nuisance factors and clustering, the toolbox provides a voxelwise map of statistical significance. Localization of the significant clusters of voxels was done using the JHU-ICBM atlas of white matter tracts (WM) [12].

As a complementary approach, we designed a network-based analysis following on previous work [13] using Principal Component Analysis (PCA) to reduce the dimensionality of the data while retaining as much variance as possible. For each patient, the number of lesioned voxels were computed based on their location on the Harvard-Oxford atlas for gray matter (GM) (https://identifiers.org/neurovault.collection:262) and on the JHU-ICBM atlas for WM. Using PCA with Promax rotation, we identified significant loadings of regions of interest (ROIs) on distinct factors, resulting in components comprising brain regions with varying lesion volumes – these components reflected patterns of lesion distribution across patients. Loading threshold was a value of 0.4 or above, and component stability was tested by comparing bootstrapping over 500 iterations. We then correlated the different factorial scores of the components to the scores of fatigue and clinical factors, with Bonferroni correction applied across all component–outcome association tests.

## Results

### Population description

Table 1 displays the descriptive statistics of the 231 participants at both time points.

**Table 1. Study population characteristics at inclusion (T1) and 3 months (T2).**

|  | T1 | T2 |
|---|---|---|
| **Sex (M/F)** | 163/68 | |
| **Age** | 57.2 (15.1) | |
| **Obesity (%)** | 10.8 | |
| **Diabetes (%)** | 10.8 | |
| **Hypertension (%)** | 50.2 | |
| **Dyslipidemia (%)** | 78.8 | |
| **Alcohol consumption (units/day)** | 0.69 (1.23) | 0.8 (0.8) |
| **HSI** | 1.95 (0.79) | 0.1 (1.1) |
| **NIHSS** | 2.1 (2.7) | 0.2 (0.4) |
| **mRS** | | 0.8 (0.5) |
| **MoCA** | 26.2 (2.7) | 26.0 (3.8) |
| **HAD-A** | 5.9 (3.1) | 5.4 (3.5) |
| **HAD-D** | 2.4 (2.3) | 3.4 (3.5) |
| **Lesion Volume (cm³)** | 3.214 (7.01) | |
| **Fazekas: Periventricular (% ≥1)** | 50.6 | |
| **Fazekas: Deep (% ≥1)** | 47.2 | |
| **Brainstem Leukoencephalopathy (%)** | 16.9 | |
| **Lobar microbleeds (% >1)** | 8.7 | |
| **Deep microbleeds (% >1)** | 9.5 | |
| **T2 Fatigue** | | |
| **Overall Fatigue** | | 45.1 (16.3) |
| **MFI General Fatigue** | | 10.4 (4.4) |
| **MFI Physical Fatigue** | | 9.4 (4.1) |
| **MFI Reduced Motivation** | | 8.1 (3.4) |
| **MFI Reduced Activity** | | 9.3 (3.7) |
| **MFI Mental Fatigue** | | 7.9 (3.8) |

Cells hold mean values and standard deviation in parentheses unless specified otherwise. HAD-A: Hospital Anxiety and Depression scale – Anxiety score, HAD-D: Hospital Anxiety and Depression scale – Depression score, HSI: Heavy Smoking Index, MFI: Multidimensional Fatigue Inventory, MoCA: Montreal Cognitive Assessment, NIHSS: National Institute of Health Stroke Scale.

Based on the clinical thresholds of the HAD scale, at inclusion, 27.2% of patients showed probable anxiety and 4.3% showed depression; at 3 months, the proportions were 25.2% and 12.6%, respectively.

Over time, there was a significant decrease in alcohol consumption (W = 10.0, p = 0.007), HSI (W = 28.0, p = 0.022), and NIHSS scores (W = 147.0, p < 0.001), while there was a significant increase in HAD Anxiety (W = 12207.0, p < 0.001) and Depression (W = 5822.5, p < 0.001).

### Associations with clinically meaningful fatigue groups

Based on the MFI cut-off scores, 20.8% of the participants endorsed a total score indicating overall fatigue. The prevalence rates by MFI subdimensions were 37.8% for General Fatigue, 53.5% for Physical Fatigue, 40.0% for Reduced Motivation, 46.1% for Reduced Activity, and 28.3% for Mental Fatigue. Descriptive statistics based on fatigue status groups are presented in Table 2.

**Table 2. Comparisons of characteristics by fatigue status group at inclusion and follow-up.**

| | General Fatigue | | Physical Fatigue | | Reduced Activity | | Reduced Motivation | | Mental Fatigue | |
|---|---|---|---|---|---|---|---|---|---|---|
| | Yes | No | Yes | No | Yes | No | Yes | No | Yes | No |
| **Sex (N: M/F)** | 53/34 | 110/33 | 86/37 | 77/30 | 74/32 | 89/35 | 61/31 | 102/36 | 46/19 | 117/48 |
| **Age** | 56.0 (15.5) | 58.0 (14.9) | 58.0 (14.4) | 56.3 (15.9) | 57.3 (15.7) | 57.1 (14.7) | 57.4 (15.8) | 57.1 (14.6) | 52.0 (15.1) | 59.3 (14.6) |
| **Obesity (%)** | 8.8 | 14.9 | 13.9 | 10.8 | 8.8 | 15.5 | 6.2 | 16.7 | 9.1 | 13.7 |
| **Diabetes (%)** | 13.8 | 9.1 | 12.2 | 9.3 | 12.3 | 9.7 | 14.1 | 9.4 | 12.3 | 10.3 |
| **Hypertension (%)** | 44.8 | 53.1 | 48.0 | 52.3 | 47.2 | 52.4 | 47.8 | 51.4 | 44.6 | 52.1 |
| **Dyslipidemia (%)** | 77 | 79.7 | 80.0 | 77.6 | 80.2 | 78.2 | 80.4 | 77.5 | 75.4 | 80.0 |
| **Alcohol (units/day)** | 0.69 (1.32) | 0.69 (1.18) | 0.72 (1.39) | 0.65 (0.97) | 0.66 (1.26) | 0.73 (1.22) | 0.80 (1.50) | 0.59 (0.96) | 0.74 (1.37) | 0.67 (1.18) |
| **HSI** | 1.9 (0.79) | 1.98 (0.78) | 1.92 (0.78) | 1.98 (0.80) | 2.00 (0.81) | 1.90 (0.77) | 2.07 (0.80) | 1.87 (0.77) | 2.02 (0.82) | 1.92 (0.77) |
| **NIHSS** | 1.75 (1.97) | 2.31 (2.98) | 1.91 (2.76) | 2.32 (2.53) | 1.95 (2.93) | 2.23 (2.41) | 1.88 (2.82) | 2.25 (2.54) | 1.58 (1.86) | 2.30 (2.89) |
| **MoCA** | 26.1 (2.88) | 26.2 (2.67) | 25.9 (2.72) | 26.5 (2.76) | 25.9 (2.70) | 26.4 (2.78) | 25.8 (3.00) | 26.4 (2.55) | 25.5 (3.10) | 26.4 (2.56) |
| **HAD-A** | 6.45 (3.01) | 5.60 (3.03) | 6.36 (2.89) | 5.42 (3.15) | 6.40 (3.06) | 5.50 (2.99) | 6.60 (3.18) | 5.47 (2.89) | 6.78 (3.21) | 5.60 (2.93) |
| **HAD-D** | 3.13 (2.42) | 1.97 (2.18) | 3.06 (2.47) | 2.47 (1.93) | 3.24 (2.54) | 1.70 (1.88) | 3.25 (2.57) | 1.85 (1.99) | 3.32 (2.76) | 2.05 (2.04) |
| **Lesion Volume (cm³)** | 2.69 (4.29) | 3.55 (8.25) | 2.37 (4.01) | 4.21 (9.29) | 2.61 (4.39) | 3.74 (8.65) | 2.42 (4.20) | 3.78 (8.37) | 2.67 (4.94) | 3.44 (7.70) |
| **Fazekas: Periventricular (%≥1)** | 50.6 | 53.6 | 50.9 | 54.4 | 47.1 | 57.1 | 53.4 | 51.9 | 44.4 | 55.6 |
| **Fazekas: Deep (%≥1)** | 43.5 | 52.2 | 50.0 | 47.6 | 46.2 | 51.3 | 48.9 | 48.9 | 44.4 | 53.6 |
| **Brainstem Leukoencephalopathy (%)** | 16.5 | 18.1 | 18.3 | 16.5 | 18.3 | 16.8 | 20.5 | 15.6 | 14.3 | 18.8 |
| **Lobar microbleeds (%>1)** | 12.0 | 7.3 | 8.5 | 10.0 | 9.7 | 8.6 | 10.3 | 8.3 | 11.7 | 8.2 |
| **Deep microbleeds (%>1)** | 9.6 | 10.3 | 10.2 | 10.0 | 9.7 | 10.3 | 12.6 | 8.3 | 11.7 | 8.9 |
| **3 month follow-up** | | | | | | | | | | |
| **Alcohol (units/day)** | 0.50 (1.01) | 0.55 (1.06) | 0.55 (1.09) | 0.52 (1.00) | 0.45 (0.91) | 0.60 (1.14) | 0.55 (1.01) | 0.52 (1.06) | 0.61 (1.09) | 0.50 (1.03) |
| **HSI** | 0.06 (0.38) | 0.07 (0.40) | 0.09 (0.48) | 0.03 (0.23) | 0.11 (0.52) | 0.02 (0.21) | 0.16 (0.61) | 0.00 (0.00) | 0.15 (0.62) | 0.03 (0.25) |
| **NIHSS** | 0.30 (0.63) | 0.14 (0.44) | 0.26 (0.60) | 0.13 (0.42) | 0.26 (0.61) | 0.15 (0.44) | 0.22 (0.55) | 0.19 (0.51) | 0.26 (0.68) | 0.18 (0.50) |
| **mRS** | 1.19 (0.71) | 0.63 (0.78) | 1.01 (0.77) | 0.63 (0.78) | 1.07 (0.79) | 0.61 (0.75) | 1.07 (0.75) | 0.68 (0.79) | 1.25 (0.68) | 0.65 (0.78) |
| **MoCA** | 25.9 (3.19) | 26.0 (4.12) | 26.0 (2.77) | 25.9 (4.71) | 25.9 (3.90) | 26.0 (3.71) | 25.6 (4.17) | 26.2 (3.51) | 25.6 (4.37) | 26.1 (3.54) |
| **HAD-A** | 7.13 (3.70) | 4.20 (2.72) | 6.18 (3.66) | 4.31 (2.85) | 6.38 (3.74) | 4.40 (2.85) | 6.61 (3.90) | 4.44 (2.77) | 7.55 (3.64) | 4.42 (2.91) |
| **HAD-D** | 5.69 (3.89) | 1.96 (2.30) | 4.71 (4.01) | 1.83 (1.88) | 5.27 (4.02) | 1.74 (1.80) | 5.62 (4.04) | 1.87 (2.01) | 5.94 (4.43) | 2.36 (2.42) |

Cells hold mean values and standard deviation in parentheses unless specified otherwise. HAD-A: Hospital Anxiety and Depression scale – Anxiety score, HAD-D: Hospital Anxiety and Depression scale – Depression score, HSI: Heavy Smoking Index, MFI: Multidimensional Fatigue Inventory, MoCA: Montreal Cognitive Assessment, NIHSS: National Institute of Health Stroke Scale

To ensure the independence of fatigue and mood measurements, we conducted an exploratory factor analysis of the HAD and the MFI (S2 Table). Six factors were obtained with intercorrelation coefficients between 0.211 and 0.614 (S3 Table). With the exception of Factor 3 and one HAD item on Factor 1, there was no overlap between the two scales.

## Categorical analysis of fatigue status

Univariate comparisons of characteristics at inclusion by fatigue status groups are presented in Table 3. Age was associated with only one MFI subdimension: the group with Mental Fatigue status had significantly younger age (mean difference: $-7 \pm 4$ years). Among stroke vascular risk factors, no significant associations with fatigue status groups were found with the exception of Fazekas Periventricular ratio and Overall Fatigue ($\chi^2 = 4.13$, $p = 0.042$).

In addition to age, confounding variables with MFI scores were the MoCA scores and the HAD Anxiety and Depression scores from T1 and T2, specific to domains of fatigue (S1 Table). Adjusting for these confounds, multivariate analyses of clinical factors by fatigue status groups are presented in Table 4.

Age and Mental Fatigue status remained significant after adjusting for MoCA scores at T1 ($F = 17.9$, $p < 0.001$) and T2 ($F = 12.51$, $p < 0.001$) as well as HAD Anxiety scores at T2 ($F = 7.136$, $p = 0.008$). Higher NIHSS scores at T2 were present in the Overall Fatigue ($F = 0.151$, $p = 0.009$) and General Fatigue ($F = 0.112$, $p = 0.021$) status groups. Adjusted for confounds, HAD Depression scores at T1 and T2, as well as higher HAD Anxiety scores at T2 were associated with having overall PSF and all fatigue subdimensions ($p < 0.001$).

The descriptive statistics by sex and their multivariate comparisons are presented in S4 and S5 Tables, respectively. A higher proportion of women were categorized in the General Fatigue subgroup (M/W%: 32.5/50.7, $p = 0.010$). The difference in

**Table 3. Univariate comparisons by post-stroke fatigue status for the characteristics at inclusion.**

|  | Overall Fatigue | General Fatigue | Physical Fatigue | Reduced Motivation | Reduced Activity | Mental Fatigue |
|---|---|---|---|---|---|---|
| **Sex** | 0.457 (0.593) | 6.372 **(0.016)** | 0.675 (0.462) | 1.444 (0.274) | 1.692 (0.228) | 0.004 (1.000) |
| **Age** | 0.173 (0.066) | 0.077 (0.331) | 0.05 (0.480) | 0.023 (0.480) | 0.001 (0.940) | 0.279 **(<0.001)** |
| **Alcohol Units/day)** | 0.061 (0.678) | 0.231 (0.160) | 0.255 (0.063) | 0.354 (0.122) | 0.356 (0.12) | 0.075 (0.621) |
| **HSI** | 0.043 (0.845) | 0.031 (0.861) | 0.001 (0.96) | 0.062 (0.49) | 0.236 (0.216) | 0.034 (0.882) |
| **Obesity** | 1.337 (0.302) | 1.382 (0.274) | 0.202 (0.663) | 3.003 (0.104) | 2.462 (0.170) | 0.548 (0.625) |
| **Diabetes** | 0.815 (0.434) | 1.153 (0.285) | 1.128 (0.293) | 0.922 (0.365) | 0.120 (0.827) | 0.215 (0.642) |
| **Hypertension** | 1.871 (0.195) | 1.722 (0.220) | 0.145 (0.789) | 0.230 (0.670) | 0.856 (0.410) | 0.935 (0.378) |
| **Dyslipidemia** | 1.330 (0.318) | 0.317 (0.617) | 0.005 (1.000) | 0.003 (1.000) | 0.011 (1.000) | 0.304 (0.591) |
| **Fazekas: Periventricular** | 4.131 **(0.042)** | 0.194 (0.659) | 0.278 (0.598) | 0.052 (0.820) | 2.245 (0.135) | 2.276 (0.132) |
| **Fazekas: Deep** | 1.333 (0.249) | 1.572 (0.21) | 0.131 (0.718) | 0.000 (0.997) | 0.579 (0.447) | 0.691 (0.406) |
| **Brainstem Leukoencephalopathy** | 0.001 (0.984) | 0.100 (0.753) | 0.128 (0.720) | 0.886 (0.347) | 0.082 (0.774) | 0.624 (0.429) |
| **Lobar microbleeds** | 0.004 (0.949) | 1.374 (0.242) | 0.133 (0.715) | 0.256 (0.613) | 0.078 (0.780) | 0.640 (0.424) |
| **Deep microbleeds** | 0.081 (0.772) | 0.024 (0.88) | 0.004 (0.947) | 1.081 (0.299) | 0.024 (0.876) | 0.240 (0.624) |

*Note:* The cells show effect size with p-value in parentheses. The comparisons for alcohol consumption and HSI were done through Mann-Whitney tests while the others through $\chi^2$ tests. Bold: significant results at $p < 0.05$. HSI: Heavy Smoking Index, MFI: Multidimensional Fatigue Inventory. The threshold for Fazekas status was greater than 0 and the threshold microbleeds was greater than 1.

Table 4. Univariate and multivariate comparisons of clinical factors by fatigue status.

| | Age | NIHSS | | MoCA | | HAD-A | | HAD-D | |
|---|---|---|---|---|---|---|---|---|---|
| | | T1 | T2 | T1 | T2 | T1 | T2 | T1 | T2 |
| **Overall Fatigue** | 0.174 (0.066) | 0.125 (0.193) | 0.151 **(0.009)** | 0.062 (0.490) | 0.01 (0.945) | 0.13 (0.186) | 35.40[e] **(<0.001)** | 17.4[c] **(<0.001)** | 106.83[c] **(<0.001)** |
| | | | | | | | 0.618[f] (0.433) | 7.58[d] **(0.006)** | 85.7[d] **(<0.001)** |
| **General Fatigue** | 0.081 (0.331) | 0.084 (0.269) | 0.112 **(0.021)** | 0.002 (0.974) | 0.071 (0.371) | 0.165 **(0.043)** | 2.22[e] (0.138) | 13.4[c] **(<0.001)** | 73.03[c] **(<0.001)** |
| | | | | | | | 4.46[f] (0.036) | 7.79[d] **(0.006)** | 34.8[d] **(<0.001)** |
| **Physical Fatigue** | 0.056 (0.480) | 0.135 (0.081) | 0.081 (0.073) | 0.152 (0.049) | 0.0887 (0.241) | 0.206 **(0.012)** | 3.02[e] (0.084) | 22.57[c] **(<0.001)** | 40.959[c] **(<0.001)** |
| | | | | | | | 0.145[f] (0.704) | 16.68[d] **(<0.001)** | 25.8[d] **(<0.001)** |
| **Reduced Motivation** | 0.022 (0.769) | 0.154 (0.051) | 0.024 (0.640) | 0.101 (0.203) | 0.102 (0.207) | 0.203 **(0.011)** | 18.13[e] **(<0.001)** | 19.65[c] **(<0.001)** | 75.617[c] **(<0.001)** |
| | | | | | | | 0.129[f] (0.720) | 15.60[d] **(<0.001)** | 56.5[d] **(<0.001)** |
| **Reduced Activity** | 0.006 (0.940) | 0.135 (0.081) | 0.083 (0.106) | 0.118 (0.139) | 0.014 (0.915) | 0.172 **(0.032)** | 14.54[e] **(<0.001)** | 25.86[c] **(<0.001)** | 67.529[c] **(<0.001)** |
| | | | | | | | 0.279[f] (0.598) | 21.09[d] **(<0.001)** | 52.1[d] **(<0.001)** |
| **Mental Fatigue** | 17.922[a] **(<0.001)** | 0.133 (0.107) | 0.070 (0.189) | 0.181 (0.031) | 0.040 (0.645) | 0.021 **(0.020)** | 39.36[e] **(<0.001)** | 16.08[c] **(<0.001)** | 54.586[c] **(<0.001)** |
| | 12.515[b] **(<0.001)** | | | | | | | | |
| | 7.136[d] **(0.008)** | | | | | | 0.279[f] (0.598) | 8.43[d] **(0.004)** | 20.5[d] **(<0.001)** |

HAD-A: Hospital Anxiety and Depression scale – Anxiety score, HAD-D: Hospital Anxiety and Depression scale – Depression score, MoCA: Montreal Cognitive Assessment, NIHSS: National Institute of Health Stroke Scale, T1: Inclusion, T2: 3-month follow up. Bold: Significant results at p<0.05.

[a]Adjusted for MoCA scores at T1.

[b]Adjusted for MoCA scores at T2.

[c]Adjusted for HAD-A scores at T1.

[d]Adjusted for HAD-A scores at T2.

[e]Adjusted for HAD-D scores at T1.

[f]Adjusted for HAD-D scores at T2.

scores between sexes remained significant after adjusting for HAD-A scores (F=4.014, p=0.046) and HAD-D scores (F=6.389, p=0.012) at T2. Examined separately, there were no significant interactions of sex with age (F=3.25, p=0.073), HAD-A (F=0.437, p=0.509), or HAD-D (F=1.55, p=0.215). No other fatigue dimension showed a significant sex-related effect.

## Dimensional analyses

The correlations between clinical factors and MFI scores are presented in Table 5 and the partial correlations adjusting for confounds are presented in Table 6.

The primary results were:

- Higher HAD Depression scores at T1 and T2, as well as higher HAD Anxiety scores at T2 were associated with scores of Overall Fatigue and all fatigue subdimensions (p<0.001).

- Age and Mental Fatigue were negatively correlated ($\rho=-0.205$, p<0.001); the association remained significant after adjusting for MoCA scores at T1 ($\rho=-0.267$, p<0.001) and T2 ($\rho=-0.226$, p<0.001) as well as HAD Anxiety scores at T2

**Table 5. Spearman correlations between MFI domain scores and the psychological, neurological and neuroimaging variables of interest.**

| | | HAD-A | HAD-D | Lesion volume | MoCA | NIHSS |
|---|---|---|---|---|---|---|
| **Overall Fatigue** | T1 | 0.219 **(<0.001)** | 0.403 **(<0.001)** | 0.028 (0.668) | −0.086 (0.192) | 0.043 (0.513) |
| | T2 | 0.480 **(<0.001)** | 0.720 **(<0.001)** | | −0.038 (0.565) | 0.135 **(0.040)** |
| **General Fatigue** | T1 | 0.164 **(0.014)** | 0.327 **(<0.001)** | 0.028 (0.676) | 0.033 (0.617) | 0.071 (0.284) |
| | T2 | 0.463 **(<0.001)** | 0.642 **(<0.001)** | | −0.019 (0.777) | 0.191 **(0.004)** |
| **Physical Fatigue** | T1 | 0.199 **(<0.001)** | 0.390 **(<0.001)** | 0.050 (0.452) | −0.133 **(0.044)** | 0.090 **(0.004)** |
| | T2 | 0.364 **(<0.001)** | 0.590 **(<0.001)** | | −0.063 (0.343) | 0.141 **(0.033)** |
| **Reduced Motivation** | T1 | 0.149 **(0.026)** | 0.338 **(<0.001)** | −0.015 (0.819) | −0.108 (0.103) | −0.021 (0.748) |
| | T2 | 0.344 **(<0.001)** | 0.638 **(<0.001)** | | −0.056 (0.402) | 0.048 (0.466) |
| **Reduced Activity** | T1 | 0.165 **(0.013)** | 0.352 **(<0.001)** | 0.046 (0.488) | −0.126 (0.056) | −0.008 (0.906) |
| | T2 | 0.336 **(<0.001)** | 0.587 **(<0.001)** | | −0.067 (0.309) | 0.110 (0.086) |
| **Mental Fatigue** | T1 | 0.212 **(0.001)** | 0.283 **(<0.001)** | −0.007 (0.921) | −0.090 (0.176) | 0.063 (0.341) |
| | T2 | 0.508 **(<0.001)** | 0.546 **(<0.001)** | | 0.012 (0.856) | 0.094 (0.155) |

HAD-A: Hospital Anxiety and Depression scale – anxiety score, HAD-D: Hospital Anxiety and Depression scale – depression score, MFI: Multidimensional Fatigue Inventory, MoCA: Montreal Cognitive Assessment, NIHSS: National Institute of Health Stroke Scale. T1: Inclusion, T2: 3 month follow up. Cells hold Spearman correlation coefficient (ρ) and p-values. Bold: Significant results at p<0.05.

**Table 6. Correlations between MFI scores and clinical factors, adjusted for potential confounding clinical variables.**

| | Age | | | HAD-A T1 | | HAD-A T2 | | | HAD-D T1 | | HAD-D T2 | |
|---|---|---|---|---|---|---|---|---|---|---|---|---|
| **Overall Fatigue** | −0.113[a] (0.089) | −0.088[b] (0.185) | −0.006[d] (0.926) | 0.107[e] (0.112) | 0.155[f] **(0.021)** | 0.520[g] **(<0.001)** | 0.499[e] **(<0.001)** | 0.136[f] **(0.040)** | 0.385[c] **(<0.001)** | 0.326[d] **(<0.001)** | 0.725[c] **(<0.001)** | 0.602[d] **(<0.001)** |
| **General Fatigue** | −0.081[a] (0.223) | −0.089[b] (0.181) | −0.024[d] (0.720) | 0.090[e] 0.184 | 0.110[f] (0.103) | 0.471[g] **(<0.001)** | 0.450[e] **(<0.001)** | 0.156[f] **(0.018)** | 0.302[c] **(<0.001)** | 0.240[d] **(<0.001)** | 0.606[c] **(<0.001)** | 0.460[d] **(<0.001)** |
| **Physical Fatigue** | −0.022[a] (0.736) | 0.001[b] (0.948) | 0.066[d] (0.317) | 0.060[e] 0.376 | 0.087[f] 0.198 | 0.387[g] **(<0.001)** | 0.347[e] **(<0.001)** | 0.041[f] 0.540 | 0.342[c] **(<0.001)** | 0.272[d] **(<0.001)** | 0.574[c] **(<0.001)** | 0.464[d] **(<0.001)** |
| **Reduced Motivation** | −0.032[a] (0.632) | −0.004[b] (0.948) | 0.075[d] (0.258) | 0.073[e] 0.280 | 0.097[f] 0.149 | 0.425[g] **(<0.001)** | 0.386[e] **(<0.001)** | 0.013[f] 0.849 | 0.339[c] **(<0.001)** | 0.285[d] **(<0.001)** | 0.650[c] **(<0.001)** | 0.567[d] **(<0.001)** |
| **Reduced Activity** | −0.070[a] (0.291) | −0.045[b] (0.500) | 0.027[d] (0.683) | 0.085[e] 0.206 | 0.117[f] 0.082 | 0.387[g] **(<0.001)** | 0.350[e] **(<0.001)** | −0.009[f] 0.890 | 0.351[c] **(<0.001)** | 0.310[d] **(<0.001)** | 0.619[c] **(<0.001)** | 0.543[d] **(<0.001)** |
| **Mental Fatigue** | −0.267[a] **(<0.001)** | −0.226[b] **(<0.001)** | −0.175[d] **(0.008)** | 0.126[e] 0.061 | 0.147[f] **(0.029)** | 0.514[g] **(<0.001)** | 0.506[e] **(<0.001)** | 0.269[f] **(<0.001)** | 0.270; **(<0.001)** | 0.193[d] **(0.003)** | 0.570[c] **(<0.001)** | 0.369[d] **(<0.001)** |

HAD-A: Hospital Anxiety and Depression – Anxiety score, HAD-D: Hospital Anxiety and Depression – Depression score, T1: inclusion, T2: 3-month follow-up. Bold: Significant results at p<0.05.

[a] Adjusted for MoCA scores at T1.

[b] Adjusted for MoCA scores at T2.

[c] Adjusted for HAD Anxiety scores at T1.

[d] Adjusted for HAD Anxiety scores at T2.

[e] Adjusted for HAD Depression scores at T1.

[f] Adjusted for HAD Depression scores at T2.

[g] Adjusted for age.

(ρ=−0.175, p=0.008). By taking three age groups – under 55 (Men/Women: 64/35), between 55 and 70 (Men/Women: 61/19), and above 70 (Men/Women: 38/14) years – Kruskal Wallis tests (S4 Table) revealed that only Mental Fatigue scores were significantly different (χ²=7.95, p=0.019) – specifically, patients under the age of 55 showed significantly higher Mental Fatigue scores (mean=8.62, SD=3.83) compared to patients aged between 55 and 70 (mean=7.60, SD=4.23).

- Lesion volume was not significantly associated with Overall Fatigue or any fatigue subdimension.

- NIHSS scores at T2 were significantly correlated with Overall Fatigue ($\rho = 0.135$, $p = 0.040$), General Fatigue ($\rho = 0.191$, $p = 0.004$), and Physical Fatigue ($\rho = 0.141$, $p = 0.033$) scores.

### Neuroimaging analyses

VLSM analysis identified a correlation between Overall Fatigue scores and lesions located in the right corona radiata and external capsule ($p = 0.024$; Fig 1). The potential confounding effect of HAD-D or HAD-A scores was ruled out since no significant clusters were associated with these scores.

Regions associated with the MFI subdimensions could not be captured by this approach because of limited overlapping lesions. Therefore, we opted for a network-based approach based on an application of principal component analysis (PCA) [13]. Since we obtained a cluster in the white matter (WM) significantly associated with Overall Fatigue scores, we based our analyses on WM regions to deepen our results as well as gray matter (GM) regions to explore what might not have been captured by the SVR-LSM analysis. Using PCA, 12 brain components of sets of WM regions and 11 brain components of sets of GM regions were extracted (Fig 2, Table 7). The variances explained by the GM and WM components were 76.9% and 63.9%, respectively. Bootstrapping revealed the first 3 GM and 2 WM components to have good stability (Tucker $\varphi > 0.8$) and weaker stability for the following components.

After adjusting for potential confounders and applying FDR correction, a white matter component (WM5) including the medial lemniscus and the cerebellar peduncles was significantly correlated with Reduced Activity ($\rho = 0.216$; $p = 0.001$) and Mental Fatigue ($\rho = 0.202$; $p = 0.002$). The Tucker $\varphi$ of the component was 0.4.

## Discussion

This study aimed to disentangle the clinical, psychological, and neuroanatomical factors associated with post-stroke fatigue. In our cohort of non-severe first-ever ischemic stroke patients, we found that approximately one in five patients experienced PSF three months after an ischemic stroke. Lesions involving motor pathways – particularly the corona radiata – and cerebro-cerebellar tracts were associated with a higher likelihood of PSF. We also observed that predisposing factors showed distinct relationships with different facets of the PSF construct, with psychological distress emerging as a prominent contributor. The use of a multidimensional tool allowed us to identify clinical, sociodemographic, and neuroradiological determinants specific to individual fatigue domains, which have been underexamined in prior studies.

In our cohort, 20.8% of patients met criteria for MFI-defined PSF – lower than the reported prevalence of 45.2% in all strokes and 36% in ischemic stroke patients [3]. This likely reflects our exclusion of patients with severe disability as well as recent or current mood disorders and the use of a multidimensional measure rather than the FSS. Consequently, the sociodemographic and clinical associations we observed were not completely aligned with the aforementioned meta-analysis by Ozkan et al. [3]: although female sex was reported to not be a significant factor for PSF, the pooled prevalence derived included studies monitoring up to 10 years post-stroke, and only 3 out of the 48 fatigue studies included featured the MFI instead of a unidimensional measure. While sex was not associated with Overall Fatigue in our study as well, women reported greater General Fatigue scores even after adjusting for stroke severity and psychological distress, suggesting sex differences in specific dimensions of fatigue perception. Although we did not observe age to be significantly associated with Overall Fatigue, younger participants endorsed higher Mental Fatigue, echoing previous findings that stroke may impose a more abrupt disruption on daily functioning in this group [4,14,15]. Stroke severity was related primarily to General and Physical Fatigue subdimensions, though our exclusion of severe cases limits conclusions about other domains. Psychological variables showed strong associations with PSF, with depression – and to a lesser extent anxiety – predicting fatigue both cross-sectionally and longitudinally [5,16]. Importantly, an exploratory factor analysis indicated that despite substantial shared variance, mood and fatigue represent partially overlapping but separate constructs

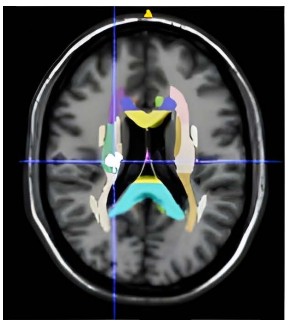
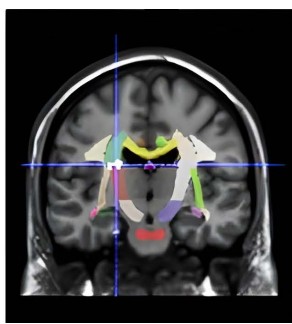
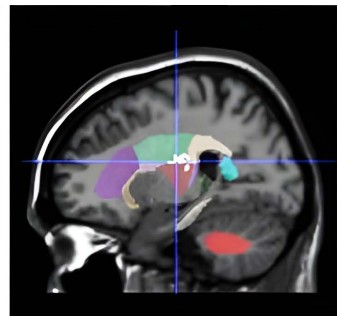

**Fig 1. Voxel-based analysis of fatigue scores.** Visualization of VLSM (voxel-based lesion-symptom mapping) results of Multidimensional Fatigue Inventory total score on a normalized template. Among the white matter tracts (colored), the significant cluster (p = 0.024) is marked in white at the crosshair.

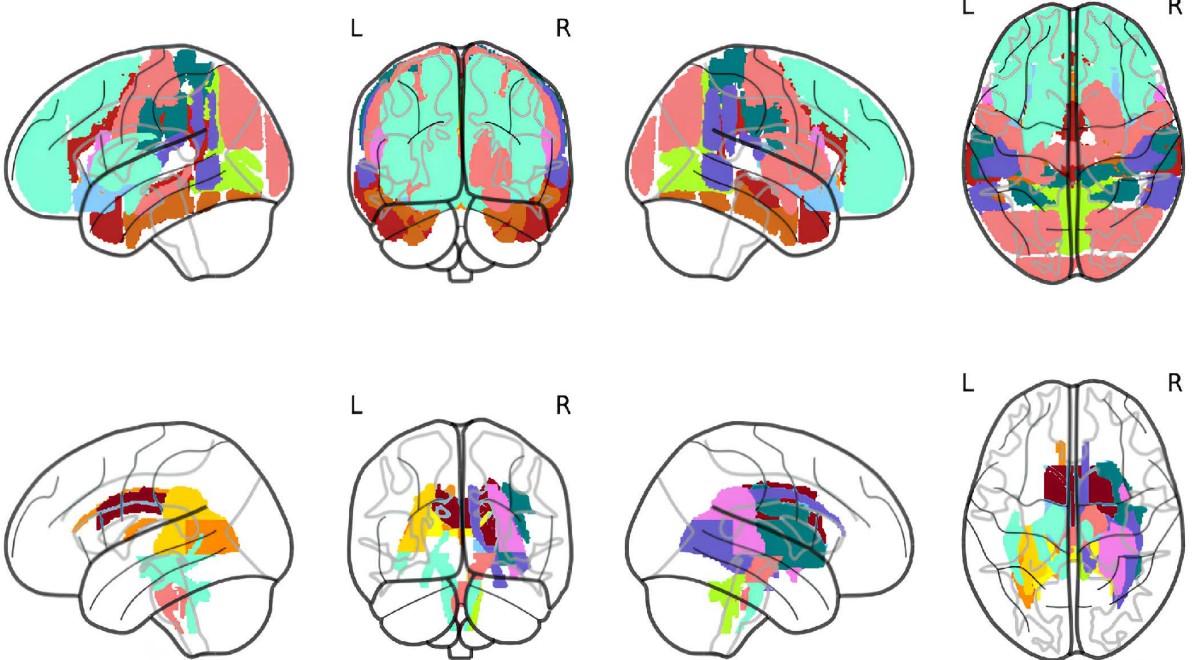

**Fig 2. Components obtained from the network-based approach. (a)** Gray matter components: GM1 (gold), GM2 (red), GM3 (chocolate), GM4 (dark orange), GM5 (light blue), GM6 (green), GM7 (teal), GM8 (purple), GM9 (coral), GM10 (pink), GM11 (turquoise). **(b)** White matter components: WM1 (yellow), WM2 (maroon), WM3 (chocolate), WM4 (dark orange), WM5 (deep red), WM6 (green), WM7 (teal), WM8 (purple), WM9 (coral), WM10 (pink), WM11 (gold), WM12 (turquoise). WM5 was significantly associated with mental fatigue scores.

rather than a single undifferentiated syndrome. Accordingly, the attenuation – but persistence – of lesion-related associations after adjustment for HAD scores likely reflects both overlapping and domain-specific symptom dimensions. Future work incorporating pre-stroke fatigue and objective fatigability measures are needed to disentangle these overlapping pathways.

Our neuroanatomical findings underscore that PSF is best understood through distributed neural mechanisms rather than focal lesion effects. Consistent with earlier work, lesion volume or microbleeds were not significantly

**Table 7. Brain regions belonging to the components obtained from the lesion network approach.**

**Grey matter networks (GM)**

| | |
|---|---|
| GM1 | Temporal Pole; Superior Temporal Gyrus, anterior division; Middle Temporal Gyrus, anterior division; Middle Temporal Gyrus, posterior division; Middle Temporal Gyrus, temporo-occipital part; Inferior Temporal Gyrus, anterior division; Inferior Temporal Gyrus, posterior division; Juxtapositional Lobule Cortex (formerly Supplementary Motor Cortex); Cingulate Gyrus, anterior division; Cingulate Gyrus, posterior division; Herschl's Gyrus |
| GM2 | Inferior Temporal Gyrus, temporo-occipital part; Parahippocampal Gyrus, anterior division; Parahippocampal Gyrus, posterior division; Lingual Gyrus; Temporal Fusiform Cortex, anterior division; Temporal Fusiform Cortex, posterior division; Occipital Fusiform Gyrus; Hippocampus (L) |
| GM3 | Subcallosal Cortex; Caudate (R); Putamen (R); Pallidum (R); Amygdala (R); Accumbens (R) |
| GM4 | Insular Cortex; Frontal Orbital Cortex; Frontal Opercular Cortex; Central Opercular Cortex; Heschl's Gyrus (includes H1 and H2); Amygdala (L) |
| GM5 | Intracalcarine Cortex; Precuneus Cortex; Cuneal Cortex; Lingual Gyrus; Supracalcarine Cortex; Occipital Pole |
| GM6 | Postcentral Gyrus; Superior Parietal Lobule; Supramarginal Gyrus, anterior division; Angular Gyrus; Lateral Occipital Cortex, superior division |
| GM7 | Middle Temporal Gyrus, temporo-occipital part; Supramarginal Gyrus, posterior division; Angular Gyrus; Parietal Opercular Cortex; Heschl's Gyrus |
| GM8 | Caudate (L); Putamen (L); Pallidum (L) |
| GM9 | Precentral Gyrus; Lateral Occipital Cortex, superior division; Occipital Pole |
| GM10 | Inferior Frontal Gyrus, pars opercularis; Frontal Opercular Cortex; Amygdala (L); Middle Frontal Gyrus |
| GM11 | Frontal Pole; Superior Frontal Gyrus; Middle Frontal Gyrus |

**White matter networks (WM)**

| | |
|---|---|
| WM1 | Genu of corpus callosum; Body of corpus callosum; Anterior limb of internal capsule (L); Posterior limb of internal capsule (L); Superior fronto-occipital fasciculus (L) |
| WM2 | Posterior limb of internal capsule (L), Anterior corona radiata (L); Superior corona radiata (L); External capsule (L); Superior longitudinal fasciculus (L) |
| WM3 | Splenium of corpus callosum, Posterior thalamic radiation (L); Cingulum (cingulate gyrus) (L) |
| WM4 | Body of corpus callosum, Anterior limb of internal capsule (R); Posterior limb of internal capsule (R); Superior corona radiata (R); External capsule (R); Superior fronto-occipital fasciculus (R) |
| WM5 | Medial lemniscus (R); Medial lemniscus (L); Inferior cerebellar peduncle (R), Superior cerebellar peduncle (R); Superior cerebellar peduncle (L) |
| WM6 | Posterior thalamic radiation (R); External capsule (R); Superior longitudinal fasciculus (R) |
| WM7 | Posterior thalamic radiation (R); Sagittal stratum (R), Cingulum (cingulate gyrus) (R); Cingulum (hippocampus) (R) |
| WM8 | Pontine crossing tract; Corticospinal tract (R); Corticospinal tract (L); Medial lemniscus (L), Cerebellar peduncle (R) |
| WM9 | Retrolenticular part of internal capsule (R); Posterior corona radiata (R), Ucinate fasciculus (R) |
| WM10 | Retrolenticular part of internal capsule (L); Posterior corona radiata (L) |
| WM11 | Middle cerebellar peduncle; Corticospinal tract (R); Inferior cerebellar peduncle (L); Superior cerebellar peduncle (L) |
| WM12 | Sagittal striatum (L); Cingulum (hippocampus) (L) |

associated with Overall Fatigue or any fatigue subdimensions [7] and we did not observe significant involvement of the thalamus or the brainstem in general – offering no support for a unitary brainstem fatigue generator model [16]. However, our SVR-LSM analysis revealed a robust association between Overall Fatigue and lesions in the right corona radiata and external capsule, reinforcing prior evidence implicating motor pathway disruption [8]. Taken together, damage in pyramidal or extrapyramidal tracts appears to be a major contributor to PSF, which may also account for the observed link between NIHSS scores and fatigue severity.

Since our voxel-based analysis was limited by low lesion overlap, we used an exploratory PCA approach to conduct network-based analysis. Although constricted by sparse data, this enabled us to form larger brain parcellations with acceptable stability. We identified associations between cerebro-cerebellar connectivity and Mental Fatigue as well as Reduced Activity subdomains, partially aligning with the aforementioned brainstem fatigue generator model. This divergence between voxel-level and network-level findings is not incompatible: while focal lesions in motor pathways appear central to Overall Fatigue, alterations in large-scale white matter networks may selectively shape cognitive-motivational aspects of fatigue [17,18]. Overall, these results follow on recent work emphasizing how a network-level approach provides a broader perspective on the neural substrates of neuropsychological dimensions in comparison to voxel-level analysis [19].

Our study has notable limitations. First, our sample included minor ischemic stroke patients without severe physical disability and no recent or current mood or anxiety disorders. However, we believe this reduces how external factors beyond the mechanisms of the stroke interact and influence the state of mental tiredness, as recognized in previous studies [20]. Moreover, since there is evidence that transient ischemic attacks can also result in long-lasting fatigue [21], restricting our cohort to a more homogeneous stroke population strengthens the validity of our findings. Nevertheless, our criteria impede any generalization of results to patients with hemorrhagic stroke or severe disabilities. Similarly, only cognitive status at inclusion and post-stroke Mental Fatigue tended to be associated, but the low level of cognitive disability of the study population precludes any firm conclusion on potential associations between post-stroke cognition and PSF. A fine-grained cognitive evaluation and a more diverse sample would be necessary to address this question. Second, the low overlap of lesion location resulting from our moderate sample size limited the SVR-LSM analyses; despite high processing power, the program could not provide convergence beyond 2000 permutations – lower than the default parameters. Third, for the network-based approach, we chose the Harvard-Oxford atlas over the MNI152 and AAL3 – which are more parcellated – due to the large number of principal components and their incoherence regarding the regions clustered from the latter. Additionally, the Harvard-Oxford atlas does not provide laterality for cortical regions, which may have impacted our findings. Finally, the present study focused on lesion characteristics and psycho-sociodemographic factors, failing to consider the influence of potential biological markers such as systemic inflammation [4].

## Conclusions

Our findings reinforce that post-stroke fatigue is a multidimensional phenomenon, underscoring the need for appropriate screening tools that capture its distinct domains. Fatigue profiles varied meaningfully across demographic groups, with age and sex influencing different dimensions, highlighting the importance of individualized clinical assessment. Consistent with prior work, depressive as well as anxiety symptoms showed strong associations with PSF. Neuroanatomically, white-matter alterations emerged as key contributors to fatigue severity, suggesting that cumulative disruption of major tracts may impair network efficiency and increase the cognitive and motor effort required for daily functioning. Overall, these findings support the need for deep phenotyping of patients in stroke follow-up to achieve more personalized management. Future work focusing on white-matter network integrity may help clarify these mechanisms.

## Supporting information

**S1 Table. Intercorrelations of clinical factors.** Spearman's correlations (ρ and p-value) between clinical factors to determine confounds for multivariate analyses. HAD-A: Hospital Anxiety and Depression–Anxiety score, HAD-D: Hospital Anxiety and Depression–Depression score, NIHSS: National Institute of Health Stroke Scale, MoCA: Montreal Cognitive Assessment. Bold: Significant results at $p < 0.05$.
(DOCX)

**S2 Table. Exploratory Factor Analysis of mood and fatigue.** The 14 items of the Hospital Anxiety and Depression (HAD) scale and the 20 items of the Mental Fatigue Inventory (MFI) are entered into an exploratory factor analysis using promax rotation and minimum residual extraction. The superscripts note the domain of the item from the two scales: HAD: [A]Anxiety and [D]Depression; MFI: [A]Reduced Activity, [G]General Fatigue, [ME]Mental Fatigue, [MO]Reduced Motivation, and [P]Physical Fatigue.
(DOCX)

**S3 Table. Pearson intercorrelations between Exploratory Factor Analysis components.**
(DOCX)

**S4 Table. Descriptive statistics and univariate comparisons between men and women at inclusion (T1) and 3 months post-stroke (T2).** Univariate comparisons were done by either Mann-Whitney U tests with rank biserial correlation ($r_{rb}$) as effect size or Chi-squared test of association ($\chi^2$); HAD-A: Hospital Anxiety and Depression scale – anxiety score, HAD-D: Hospital Anxiety and Depression scale – depression score, HSI: Heavy Smoking Index, MFI: Multidimensional Fatigue Inventory, MoCA: Montreal Cognitive Assessment, mRS: Modified Rankin Scale, NIHSS: National Institute of Health Stroke Scale. Bold: significant results at $p < 0.05$.
(DOCX)

**S5 Table. Multivariate comparison of MFI scores by sex adjusted for potential confounding clinical factors.** Cells show F-statistic from Quade tests and p-value in parentheses. Bold: Significant results at $p < 0.05$. [a]adjusted for HAD Anxiety scores at T2. [b]adjusted for HAD Depression scores at 3 months post-stroke.
(DOCX)

**S6 Table. Age-stratified comparisons.** Kruskal-Wallis tests between the three age groups: under 55, between 55–70 and above 70 years. HAD-A: Hospital Anxiety and Depression–Anxiety score, HAD-D: Hospital Anxiety and Depression–Depression score, MFI: Multidimensional Fatigue Inventory. Bold: Significant results at $p < 0.05$.
(DOCX)

## Acknowledgments

The authors would like to acknowledge all the participants, Sylvain Ledure for his assistance in conducting the study, and the Clinical Epidemiology Unit of Bordeaux University hospital (USMR), in particular Antoine Bénard and Florence Allais, for their valuable support in the data collection and storage process.

## Author contributions

**Conceptualization:** Igor Sibon, Sylvie Berthoz.

**Data curation:** Suhrit Duttagupta.

**Formal analysis:** Suhrit Duttagupta, Lutzi Castano.

**Funding acquisition:** Igor Sibon, Sylvie Berthoz.

**Investigation:** Sylvie Berthoz.

**Methodology:** Suhrit Duttagupta.

**Project administration:** Igor Sibon, Sylvie Berthoz.

**Resources:** Igor Sibon, Sylvie Berthoz.

**Software:** Suhrit Duttagupta, Lutzi Castano, Sandra Chanraud.

**Supervision:** Sandra Chanraud, Igor Sibon, Sylvie Berthoz.

**Writing – original draft:** Suhrit Duttagupta, Lutzi Castano.

**Writing – review & editing:** Igor Sibon, Sylvie Berthoz.

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
