## [Decision Letter · Decision Letter 0]

30 Jan 2026

Dear Dr. Duttagupta,

Thank you for submitting your manuscript to PLOS ONE. After careful consideration, we feel that it has merit but does not fully meet PLOS ONE’s publication criteria as it currently stands. Therefore, we invite you to submit a revised version of the manuscript that addresses the points raised during the review process.

Based on the reviewers' suggestions, the paper needs a major revision. The reviewers' comments can be found below.

We look forward to receiving your revised manuscript.

Kind regards,

Tanja Grubić Kezele, Ph.D., M.D.

Academic Editor

PLOS One

Journal Requirements:

2. Please describe in your methods section how capacity to provide consent was determined for the participants in this study. Please also state whether your ethics committee or IRB approved this consent procedure. If you did not assess capacity to consent please briefly outline why this was not necessary in this case.

“Funders of this work were the French ministry of Health (reference number: PHRC-17-0377) and the French National Research Agency (ANR), under the France 2030 program (reference number: ANR-23-IAHU-0001). Funders of the PhD students involved in this work are the Doctoral School of Bordeaux Neurocampus Graduate Program (SD) and the EPHE-PSL Research University (LC).”

4. In the online submission form you indicate that your data is not available for proprietary reasons and have provided a contact point for accessing this data. Please note that your current contact point is a co-author on this manuscript. According to our Data Policy, the contact point must not be an author on the manuscript and must be an institutional contact, ideally not an individual. Please revise your data statement to a non-author institutional point of contact, such as a data access or ethics committee, and send this to us via return email. Please also include contact information for the third party organization, and please include the full citation of where the data can be found.

5. Please amend either the title on the online submission form (via Edit Submission) or the title in the manuscript so that they are identical.

Reviewers' comments:

Reviewer's Responses to Questions

**Comments to the Author**

1. Is the manuscript technically sound, and do the data support the conclusions?

Reviewer #1: Partly

Reviewer #2: No

2. Has the statistical analysis been performed appropriately and rigorously?

Reviewer #1: Yes

Reviewer #2: No

3. Have the authors made all data underlying the findings in their manuscript fully available?

Reviewer #1: Yes

Reviewer #2: Yes

4. Is the manuscript presented in an intelligible fashion and written in standard English?

Reviewer #1: Yes

Reviewer #2: Yes

Reviewer #1: The research uses a variety of statistical techniques to examine the risk factors for PSF including parametric and non parametric. Such are the Pearson correlation, Quade, MFI score, neuro imaging (SVR-LSM) and the complementary PCA network based analysis. The Results section gives a rather fragmented summary of the factors or variables which may be the major players at the times of interest and summarized in about 7 tables. The precise statistical relationship among these variables and factors is not totally clear. For example is there a collinearity issue in the multivariate approach? Also synthesizing this information can be very challenging or cumbersome. How can this information be summarized or combined in a factor approach or in some other data reduction format? Some strategies have used a score function to give some sort of a hierarchy of summary risk level of the factors combined for the endpoint of interest (for PSF for example) using the logistic model or a time dependent model (such as Cox regression). This gives a risk profile of the information and subjects can be classified as to their level of risk based on their total score.

This paper represents a lot of work by the investigators and they do note in their conclusion that their findings

reinforce that post-stroke fatigue is a multidimensional phenomenon, underscoring the need for appropriate screening tools that capture its distinct domains. Can their domains be combined as noted above to derive a more interpretable risk profile.

Reviewer #2: The manuscript requires major revision. It is not technically sound (unrepresentative sample, conflated fatigue-depression constructs, meta-analytic inconsistency) nor statistically rigorous (lacks subgroup/factor analyses, unclear FDR scope/stability). Data availability and English expression are adequate. Address the issues as recommended to improve technical robustness and statistical validity.

**Do you want your identity to be public for this peer review?** For information about this choice, including consent withdrawal, please see our Privacy Policy

Reviewer #1: No

Reviewer #2: **Yes:** Tao Jiang

---

## [Author Response · Author response to Decision Letter 1]

25 Feb 2026

Editor Comment 1: Please ensure that your manuscript meets PLOS ONE's style requirements, including those for file naming. The PLOS ONE style templates can be found at https://journals.plos.org/plosone/s/file?id=wjVg/PLOSOne_formatting_sample_main_body.pdf and https://journals.plos.org/plosone/s/file?id=ba62/PLOSOne_formatting_sample_title_authors_affiliations.pdf

Response: We have now modified the text and the file names in the appropriate format. To ensure the ‘Track Changes’ comments have sufficient clarity, we pre-accepted functional text changes such as font size or bold text.

Editor Comment 2: Please describe in your methods section how capacity to provide consent was determined for the participants in this study. Please also state whether your ethics committee or IRB approved this consent procedure. If you did not assess capacity to consent please briefly outline why this was not necessary in this case.

Response: We have now specified in the methods (line 112-113) that capacity to consent was provided by the participant’s ability to read and write, as approved by the local ethics committee.

Editor Comment 3: Thank you for stating the following financial disclosure:

“Funders of this work were the French ministry of Health (reference number: PHRC-17-0377) and the French National Research Agency (ANR), under the France 2030 program (reference number: ANR-23-IAHU-0001). Funders of the PhD students involved in this work are the Doctoral School of Bordeaux Neurocampus Graduate Program (SD) and the EPHE-PSL Research University (LC).”

Response: We have now included this statement in our cover letter.

Editor Comment 4: In the online submission form you indicate that your data is not available for proprietary reasons and have provided a contact point for accessing this data. Please note that your current contact point is a co-author on this manuscript. According to our Data Policy, the contact point must not be an author on the manuscript and must be an institutional contact, ideally not an individual. Please revise your data statement to a non-author institutional point of contact, such as a data access or ethics committee, and send this to us via return email. Please also include contact information for the third party organization, and please include the full citation of where the data can be found.

Response: We have mentioned in the online form that the contact point will be the Methodological Support for Clinical and Epidemiological Research (USMR) at Bordeaux University hospital. Since there is no institutional email, the head of the unit, Dr. Antoine Benard, will be the point of reference.

Editor Comment 5: Please amend either the title on the online submission form (via Edit Submission) or the title in the manuscript so that they are identical.

Response: We have now corrected this error, in addition to revising the title as per the reviewer comments.

Editor Comment 6: Please include captions for your Supporting Information files at the end of your manuscript, and update any in-text citations to match accordingly. Please see our Supporting Information guidelines for more information: http://journals.plos.org/plosone/s/supporting-information.

Response: We have now added the captions and updated the citations correctly.

Editor Comment 7: If the reviewer comments include a recommendation to cite specific previously published works, please review and evaluate these publications to determine whether they are relevant and should be cited. There is no requirement to cite these works unless the editor has indicated otherwise.

Response: There were no recommendations for citing previous works.

Responses to the reviewers for Manuscript PONE-D-26-00845

Reviewer #1 Comment 1: The research uses a variety of statistical techniques to examine the risk factors for PSF including parametric and non parametric. Such are the Pearson correlation, Quade, MFI score, neuro imaging (SVR-LSM) and the complementary PCA network based analysis.

The Results section gives a rather fragmented summary of the factors or variables which may be the major players at the times of interest and summarized in about 7 tables.

Response: We acknowledge that by using not only a multilevel approach (psychometric and neuroimaging), and both a categorical and a dimensional approach to post-stroke fatigue, reorganizing the Results section by analytical domains would improve the Ms presentation and quality. We have also modified the Methods section accordingly. The presentation of the Results section is now organized as follows:

Descriptive Statistics

Categorical Analyses

Dimensional Analyses

Neuroimaging Analyses

In addition, we have decided to keep in the main text only the Tables related to the main effects and thus moved both sex-specific tables as supplementary material (S4 and S5 Tables).

Reviewer #1 Comment 2: The precise statistical relationship among the variables and factors is not totally clear. For example is there a collinearity issue in the multivariate approach? Also synthesizing this information can be very challenging or cumbersome. How can this information be summarized or combined in a factor approach or in some other data reduction format? Some strategies have used a score function to give some sort of a hierarchy of summary risk level of the factors combined for the endpoint of interest (for PSF for example) using the logistic model or a time dependent model (such as Cox regression). This gives a risk profile of the information and subjects can be classified as to their level of risk based on their total score.

Response: We thank the reviewer for this thoughtful comment regarding the precise statistical relationship among the variables.

We agree that data-reduction approaches and composite risk scores can be useful in prediction-focused studies. However, our primary objective was to examine the contributions of socio-demographic, psychological, and neuroanatomical factors to different dimensions of post-stroke fatigue, rather than to develop a predictive or risk-scoring model. Given the heterogeneity of data modalities in the present work (clinical, psychological, and neuroimaging variables) and the exploratory, mechanistic aim of our analyses, we believe that combining these domains into a single latent factor or summary score would risk obscuring domain-specific effects and biological interpretability. Moreover, fatigue was assessed only once (3 months post-stroke), and regrettably the study design is not suited for longitudinal risk modeling. Therefore, predictive modeling approaches such as time-to-event analyses or composite risk profiling fall outside the scope of the present investigation.

Regarding potential multicollinearity, the univariate correlations between confound variables (S1 Table) and the correlations between MFI scores and clinical variables in Table 5 indicate that with the exception of certain fatigue and HAD score associations, the correlation coefficients fall under 0.4, which would be considered “weak” according to recommended guidelines (Schober et al., 2018). For the analyses using the MFI and HAD scores together, we conducted linear regressions in the same format as the ranked ANCOVA and found the VIF values to be around 1, which indicates negligible collinearity. In line with the Reviewer’s comment, this information has been added in the Revised Ms (lines 155 and 156).

In addition, please refer to Reviewer 2’s Comment 2: interaction terms (e.g., sex×age, sex×HAD) were formally tested and found to be non-significant (lines 260–262), supporting the independence of major effects.

Reference: Schober P, Boer C, Schwarte LA. Correlation coefficients: appropriate use and interpretation. Anesthesia & analgesia. 2018;126(5):1763-8.

Reviewer #2 Comment 1: Compromised Sample Representativeness

The exclusion of patients with "any mood or anxiety disorder in the past 6 months" represents a stringent design choice. While this approach is justified from the perspective of reducing confounding factors, it makes the study cohort highly unrepresentative of the general post-stroke population (post-stroke depression and anxiety prevalence: 25-40%). Consequently, the prevalence of post-stroke fatigue (PSF) in this cohort (20.8%) is substantially lower than the reported population-level estimates (45.2%), which raises questions about the generalizability of the findings. More fundamentally, this exclusion strategy changes the research question from "what causes PSF?" to "what causes PSF in a specific subgroup of mild stroke patients without psychiatric comorbidities?"—a notably narrower inquiry. Recommendation: The authors should more prominently highlight the specificity of their sample in both the abstract and the discussion. A sentence such as the following would enhance transparency: "This cohort is restricted to patients with mild-to-moderate ischemic stroke and no prior mood or anxiety disorders and may not be generalizable to patients with concurrent psychiatric conditions."

Response: We thank the reviewer for this important comment. We fully agree that our exclusion criterion limits the generalizability of our findings and indeed this choice was made to reduce confounding factors that were found to impact the range of prevalence of post-stroke depression and anxiety. We have now noted this design choice in the Abstract as well as in the Discussion (lines 407-408) to highlight the specificity of our sample, emphasizing that our cohort consisted of patients with first-ever mild ischemic stroke without recent depressive or anxiety disorders and have explicitly stated that our findings may not generalize to patients with more severe strokes or with concurrent psychiatric conditions.

We also clarified in the Methods (lines 100-102) and the Discussion (paragraph with lines 349-358, limitations section lines 412-414) that this design choice was intended to reduce heterogeneity and confounding effects, allowing us to examine lesion-related correlates of fatigue in a relatively homogeneous subgroup. Finally, we have also altered our title to indicate this specificity.

Reviewer #2 Comment 2: Inconsistency with Existing Meta-Analytic Evidence

A striking finding is that the authors report higher rates of general fatigue in female patients (M/W: 32.5%/50.7%, p=0.010), which directly contradicts a recent meta-analysis by Ozkan et al. (2024) concluding that female sex is not a significant PSF risk factor. However, the current discussion only briefly mentions "variability in sample characteristics" to dismiss this discrepancy, which is an insufficient explanation. The tension remains unresolved. Recommendation: Conduct stratified subgroup analyses by sex and age groups (<55, 55-70, >70 years). Report the statistical significance of sex×age interaction terms (and other relevant interactions such as sex×HAD score).

Response: We agree with the reviewer on the need to further address this important discrepancy. We want to specify that this finding was for the specific MFI domain of General Fatigue and not the overall fatigue score (i.e MFI Total score), and that Ozkan et al. (2024) did not analyze the subdomains of the fatigue construct. This is now specified in the revised Discussion (lines 359-362). Additionally, we have noted in the revised Discussion that this discrepancy may be due to 2 factors: (1) Only 3 out of the 48 fatigue studies included in the aforementioned meta-analysis used the MFI over a unidimensional measure, and (2) while our study evaluated PSF only at 3 months post-stroke, the meta-analysis pooled fatigue prevalences from up to 10 years post-stroke (lines 364-367).

The sex-stratified comparisons had been conducted in the previous version of the Ms (previous Tables 3 and 5 - now shifted to S4 and S5 Tables). In line with the Reviewer’s recommendation, the statistical significance of the sex-age and sex-HAD scores are now reported in the revived Results section (lines 260-262), and we have now added the age-stratified comparisons (<55, 55-70, >70 years) in the Supplementary Table S6.

Reviewer #2 Comment 3. Conceptual Conflation of Fatigue and Depression Symptomatology

Table 4 reveals a correlation of ρ=0.720 (p<0.001) between the Hospital Anxiety and Depression Scale-Depression (HAD-D) and the Multidimensional Fatigue Inventory (MFI) total fatigue score—a remarkably high association suggesting substantial symptom overlap. Although the authors correctly note in the introduction that fatigue and depression share overlapping experiential dimensions (e.g., energy deficit, psychomotor slowing; lines 61-65), the study does not statistically separate these constructs. The critical unresolved question is: What does this 0.72 correlation represent? Two independent but closely related symptom constructs? Different manifestations of a single underlying factor (measurement confounding)? A shared neurobiological mechanism? When the authors adjust for HAD scores (Table 6), the associations with MFI weaken but are not eliminated, suggesting potential measurement-level confounding rather than a true independent effect. This significantly undermines the interpretability of claims regarding the relative contributions of psychological versus neurobiological factors to PSF. Recommendation: Conduct exploratory factor analysis or structural equation modeling to decompose MFI and HAD into common factors and unique factors. Reanalyze key associations using these decomposed factors rather than total scores. Explicitly discuss in the results and discussion the conceptual and statistical implications of symptom overlap, making it clear whether the reported associations reflect true independence or measurement-level confounding.

Response: We thank the reviewer for giving us the opportunity to emphasize more on this crucial issue. To address the possibility of measurement-level confounding and shared latent structure, we conducted an exploratory factor analysis including all 14 HAD items and 20 MFI items, using oblique (promax) rotation. These analyses are now presented in S2 and S3 Tables and described in the Results (lines 219–222).

The factor solution revealed that, with the exception of one factor, HAD and MFI items loaded predominantly on separate latent dimensions, indicating largely distinct underlying constructs. It must be acknowledged that the factors have moderate to high intercorrelations (magnitudes between 0.296 - 0.614) and the aforementioned factor with overlapping measurements has items of anxiety, depression, and separate fatigue domains. Overall, this reflects substantial shared variance between the two related but separable symptom domains, rather than a single undifferentiated factor. To further clarify this point in the revised Ms, we now emphasize in the Discussion (lines 377–379) that fatigue and mood symptoms - as measured by the MFI and HAD respectively - share overlapping experiential features and may be driven in part by common mechanisms, but they also retain domain-specific variance captured by each scale.

Reviewer #2 Comment 4. PCA network analysis: insufficient specification of the multiple-testing family, FDR scope, and stability

For the PCA-based network analysis, the manuscript does not adequately specify the testing family or the exact scope at which the false discovery rate (FDR) correction is applied. Given the modest effect sizes, transparency about the correction scope and component stability is essential. Recommendation: Clearly define the family of tests subjected to FDR control (e.g., components×outcomes), report the variance explained and loading thresholds, and add a basic stability assessment (bootstrap or split-half). The Discussion should explicitly present these PCA results as exploratory/hypothesis-generating unless robust stability evidence is provided.

---

## [Decision Letter · Decision Letter 1]

4 Mar 2026

Identifying clinico-radiological determinants of post-stroke fatigue 3 months post-stroke in a French hospital-based cohort of non-severe stroke patients without psychiatric comorbidites

PONE-D-26-00845R1

Dear Dr. Duttagupta,

We’re pleased to inform you that your manuscript has been judged scientifically suitable for publication and will be formally accepted for publication once it meets all outstanding technical requirements.

Kind regards,

Tanja Grubić Kezele, Ph.D., M.D.

Academic Editor

PLOS One

Additional Editor Comments (optional):

Reviewers' comments:

Reviewer's Responses to Questions

**Comments to the Author**

Reviewer #1: All comments have been addressed

2. Is the manuscript technically sound, and do the data support the conclusions?

Reviewer #1: (No Response)

3. Has the statistical analysis been performed appropriately and rigorously?

Reviewer #1: (No Response)

4. Have the authors made all data underlying the findings in their manuscript fully available?

Reviewer #1: (No Response)

5. Is the manuscript presented in an intelligible fashion and written in standard English?

Reviewer #1: (No Response)

Reviewer #1: (No Response)

**Do you want your identity to be public for this peer review?** For information about this choice, including consent withdrawal, please see our Privacy Policy

Reviewer #1: No

---

## [Editor Report · Acceptance letter]

PONE-D-26-00845R1

PLOS One

Dear Dr. Duttagupta,

I'm pleased to inform you that your manuscript has been deemed suitable for publication in PLOS One. Congratulations! Your manuscript is now being handed over to our production team.

Kind regards,

on behalf of

Prof. dr. Tanja Grubić Kezele

Academic Editor

PLOS One